# What Are the Determinants of the Quality of Systematic Reviews in the International Journals of Occupational Medicine? A Methodological Study Review of Published Literature

**DOI:** 10.3390/ijerph20021644

**Published:** 2023-01-16

**Authors:** Giuseppe La Torre, Remigio Bova, Rosario Andrea Cocchiara, Cristina Sestili, Anna Tagliaferri, Simona Maggiacomo, Camilla Foschi, William Zomparelli, Maria Vittoria Manai, David Shaholli, Vanessa India Barletta, Luca Moretti, Francesca Vezza, Alice Mannocci

**Affiliations:** 1Department of Public Health and Infectious Diseases, Sapienza University of Rome, 00185 Rome, Italy; 2Faculty of Economics, Universitas Mercatorum, 00185 Rome, Italy

**Keywords:** quality assessment, public health, occupational health, PRISMA statement, AMSTAR-2 score

## Abstract

Objective: The aim of this study was to evaluate the methodological quality of systematic reviews published in occupational medicine journals from 2014 to 2021. Methods: Papers edited between 2014 and 2021 in the 14 open access journals with the highest impact were assessed for their quality. Studies were included if they were systematic reviews and meta-analyses, and if they were published in English. Results: The study included 335 studies. Among these, 149 were meta-analyses and 186 were systematic reviews. The values of the AMSTAR-2 score range between three and fourteen with a mean value of 9.85 (SD = 2.37). The factors that significantly and directly associate to a higher AMSTAR-2 score were impact factor (*p* = 0.003), number of consulted research databases (*p* = 0.011), declaration of PRISMA statement (*p* = 0.003), year of publication (*p* < 0.001) and performing a meta-analysis (*p* < 0.001).The R² values from the multivariate analysis showed that the AMSTAR-2 score could be predicted by the inclusion of these parameters by up to 23%. Conclusions: This study suggests a quality assessment methodology that could help readers in a fast identification of good systematic reviews or meta-analyses. Future studies should analyze more journals without applying language restrictions and consider a wider range of years of publication in order to give a more robust evidence for results.

## 1. Introduction

The use of reporting guidelines is uncommon in Public Health (PH) research, and no specific guideline application has been recommended at the European level [1,2,3].

The first formal assessment of Systematic Reviews (SRs) in medicine was performed by Cynthia Mulrow, who identified several poor reporting practices of 50 medical review articles published between June 1985 and June 1986 [4]. In the last 10 years, several reporting guidelines have been developed for reporting SRs. In 1999, an international group of 30 epidemiologists, clinicians, statisticians, editors and researchers developed a reporting guideline for meta-analyses of randomized trials—the QUOROM (QUality Of Reporting Of Meta-analyses) Statement [5]. In 2005, a meeting was convened to update QUOROM to address several conceptual and practical advances in the methodology of SRs and to help overcome several shortcomings identified in an audit of SRs [6]. The guideline was renamed the PRISMA (Preferred Reporting Items for Systematic reviews and Meta-Analyses) Statement, and it was published in 2009 [7]. It was accompanied by an explanation and elaboration document, which provided detailed guidance for each of the 27 items included and examples of exemplar reporting [8]. Since its publication, seven extensions to the PRISMA Statement have been developed to facilitate reporting of different types or aspects of SRs [9,10,11,12,13,14,15,16,17,18,19,20]. These guidelines present a sequence of indications that should enhance the understanding and interpretation of studies which may be difficult for the reader, such as relevant information which may not be adequately described or perhaps poorly presented [8,21]. 

In recent years, different assessment tools have been developed to assess the methodological quality of SRs, and among these tools, the ones used most commonly are AMSTAR, ROBIS and SIGN. The “Assessing the Methodological Quality of Systematic Reviews” (AMSTAR) was created in 2007 [22] and updated (AMSTAR-2) [23] in 2017 for facilitating the development of high-quality reviews by focusing on their methodological quality. AMSTAR 2 is one of the most widely used instruments for enabling rapid and reproducible assessments of the quality of systematic reviews in both randomised and non-randomised studies of healthcare interventions. 

ROBIS is used to assess the risk of bias in SRs [24]. This tool is based on three phases, the first one being optional, and the other two covering 4 four domains each, i.e., identifying and appraising study eligibility criteria, identification and selection of studies, data collection and study appraisal, and synthesis and findings, for the evaluation of the overall risk of bias.

SIGN has been developed by the Scottish Intercollegiate Guidelines Network for systematic review checklist; it is a 12-item checklist developed on the basis of the AMSTAR checklist [25]. The objective of the tools developed by the SIGN is to improve the quality of health care for patients by reducing variation in practice and outcome, through the development and dissemination of national clinical guidelines containing recommendations for effective practice based on current evidence.

The aim of this study was to evaluate the methodological quality of the systematic reviews published in occupational medicine journals between 2014 and 2021 and to assess related factors of the scientific publication that are associated with the methodological quality assessed by the AMSTAR-2. 

## 2. Materials and Methods

### 2.1. Eligibility Criteria

This research study included all the systematic reviews and meta-analyses published in the 14 occupational medicine journals with the highest impact from January 2014 to December 2021. All the papers written in a different period were excluded, as well as randomized control trials, narrative reviews, surveys, reports, protocols and pilot studies. All the articles not published in English language were excluded. The typology of the study (e.g., review or other) had to be mentioned in the title or abstract.

### 2.2. Information Sources

This study is a methodological study review carried out by checking for publications in the period 2014–2021 of open access occupational medicine journals with the highest impact.

Only English papers were selected.

The following occupational medicine journals were analyzed:(1)American Journal of Industrial Medicine;(2)Annals of Occupational Hygiene;(3)Archives of Environmental and Occupational Health;(4)Environmental Health Perspectives;(5)International Journal of Environmental Research and Public Health;(6)International Journal of Occupational and Environmental Health;(7)International Journal of Occupational Medicine and Environmental Health;(8)International Journal of Occupational Safety and Ergonomics;(9)Journal of Occupational and Environmental Hygiene;(10)Journal of Occupational and Environmental Medicine;(11)Journal of Occupational Medicine and Toxicology;(12)Occupational and Environmental Medicine;(13)Occupational Medicine;(14)Scandinavian Journal of Work, Environment & Health.

Each issue of the 14 journals, excluding supplements, was examined.

### 2.3. Selection of Sources of Evidence

The websites of the journals were consulted to evaluate the instructions for the authors. Literature search and data extraction were performed by two authors independently. Only systematic review and meta-analysis study designs were selected. 

### 2.4. Data Items

The choice of items to be selected was based on the results of previous research in different health fields. Pauletto et al. made a critical appraisal of systematic reviews of intervention in dentistry, using as explanatory variables of methodological quality items such as publication year, continent and journal impact factor [26].

Cheung et al. assessed the methodological quality of systematic reviews on Chinese herbal medicine, finding higher levels in SRs conducted by more authors and published in higher impact factor journals [27].

McGregor et al. found that the quality of meta-analyses of non-opioid, pharmacological, perioperative interventions for chronic postsurgical pain is associated with time of publication and journal impact factor [28].

Yuan et al., assessing Breast Reconstruction Reviews, studied the association between AMSTAR score and impact factor, number of citations, number of studies and adherence to the PRISMA statement [29]. 

Chow et al., making the quality appraisal of systematic reviews on methods of labor induction, studied the association between study quality and number of citations, journal impact factor and publication year [30].

Yuan et al. made the assessment of the Quality of Systematic Reviews and Meta-Analyses About Breast Augmentation by studying the association between AMSTAR score and journal impact factor, number of citations, year of publication, number of included studies and adherence to PRISMA guidelines [31].

The following variables were collected:Name of the first author;Name of the journal;Title of the article;Year of publication;Adherence to PRISMA Checklist made by the authors (yes/no);Declaration by the authors of the use of PRISMA Checklist (yes/no);Journal impact factor;Numbers of authors;Nationality of the first author;Number of investigated databases in the review;Multi-country research group;Total number of articles included in the systematic review;Implementation of meta-analysis;Total AMSTAR-2 score: “yes” (all the criteria were met), “no” (none of the criteria were met), “partial yes” (not all the criteria were met).AMSTAR-2 checklist.

### 2.5. AMSTAR-2 Checklist

All the authors were involved in the assessment, and two researchers scored the same paper. If differences in the assessment were present, a third reviewer was involved. 

Even though AMSTAR2 is an instrument that explicitly states that it is not designed to generate an overall score, it can be useful for rating the overall confidence in the results of the review. Hence, we decided to calculate an overall score and use this as the dependent variable in the statistical analysis, as executed by several authors [32,33,34,35].

The AMSTAR-2 checklist answers to the following 16 questions:-Did the research questions and inclusion criteria for the review include the components of PICO?-Did the report of the review contain an explicit statement that the review methods were established prior to the conduct of the review and did the report justify any significant deviations from the protocol?-Did the review authors explain their selection of the study design for inclusion in the review?-Did the review authors use a comprehensive literature search strategy?-Did the review authors perform study selection in duplicate?-Did the review authors perform data extraction in duplicate?-Did the review authors provide a list of excluded studies and justify the exclusions?-Did the review authors describe the included studies in adequate detail?-Did the review authors use a satisfactory technique for assessing the risk of bias (RoB) in individual studies that were included in the review?-Did the review authors report on the sources of funding for the studies included in the review?-If a meta-analysis was performed, did the review authors use appropriate methods for the statistical combination of results?-If a meta-analysis was performed, did the review authors assess the potential impact of RoB in individual studies on the results of the meta-analysis or other evidence synthesis?-Did the review authors account for RoB in primary studies when interpreting/discussing the results of the review?-Did the review authors provide a satisfactory explanation for, and discussion of, any heterogeneity observed in the results of the review?-If they performed quantitative synthesis, did the review authors carry out an adequate investigation of publication bias (small study bias) and discuss its likely impact on the results of the review?-Did the review authors report any potential sources of conflict of interest, including any funding they received for conducting the review?

### 2.6. Statistical Analysis

The description of results included qualitative variable description reporting the percentage of articles that declared and used the PRISMA statement, the AMSTAR-2 score frequency and distribution, a univariate and bivariate analysis of AMSTAR-2 score vs. qualitative and quantitative variables and a multivariate regression model that analyzed the influence of the variables on the AMSTAR-2 score.

A descriptive analysis was performed in order to summarize the characteristics of the articles. The AMSTAR-2 score was described with mean and standard deviation (SD), while the qualitative items (multi-country research team, PRISMA declared, PRISMA applied, meta-analysis) were described by frequencies and percentages. A univariate analysis was performed in order to assess the association between the AMSTAR-2 score and the collected variables from the form sheet. The correlation analysis was carried out to assess the relationship between two quantitative variables. Pearson’s r coefficient (also known as the linear correlation coefficient), used for variables measured by interval scales or relation scales, was reported. A multivariate analysis was conducted with a backward stepwise elimination procedure of non-significant variables generating a minimal model. The inclusion criterion for the covariates in the models was a significance level of <0.20 from the univariate analysis. 

The goodness of fit of the models was assessed using the R^2^.

The level of significance was set at *p* < 0.05. The statistical analysis was performed using SPSS (Statistical Package for Social Sciences) software, version 27. 

## 3. Results

Two different authors independently identified 1017 reviews (systematic reviews, narrative reviews, protocols, surveys and pilot studies) published in the selected journals between January 2014 and December 2021. From these 1017 potentially relevant articles, 553 were excluded since they were not systematic reviews and 123 were out of scope; 341 were selected. Among these, six articles were excluded because they were protocols. The number of articles included was 335, out of which 149 were meta-analyses and 186 were systematic reviews. The PRISMA flow diagram represented in Figure 1 summarizes this selection process.

Concerning the AMSTAR-2 score, Figure 2 shows the distribution of AMSTAR-2 score frequencies among the included studies. Values range between three and fourteen with a mean value of 9.85 (SD = 2.38) and a median value of 10. In Figure 3, the percentages of systematic reviews not reporting the 16 items of the AMSTAR-2 are shown, indicating the highest non-adherence for the items funding (74%), excluded study list (70.2%) and impact of risk of bias (52.5%).

The occupational health areas are described in Table 1. Half of the systematic reviews are represented by papers on toxicology (13.2%), health promotion and intervention (11%), environmental exposure and climate changes (10.7%), mental and neurological health (9.6%) and musculoskeletal disorders (9.6%).

### 3.1. Univariate and Bivariate Analysis

Table 2 shows the univariate analysis of AMSTAR-2 score regarding the qualitative variables taken in consideration for this study. The resulting AMSTAR-2 score was significantly associated with all qualitative variables such as multi-country research team, PRISMA statement application, PRISMA statement declared and meta-analysis. In fact, having a multi-country research team, having a PRISMA statement declared in the text, having applied PRISMA and having a meta-analysis increased the AMSTAR-2 score.

The correlation analysis is shown in Table 3. The correlation analysis studied the relationship between quantitative variables. A significant correlation with the AMSTAR-2 score was found for impact factor (*p* = 0.001), year of publication (*p* < 0.001), increasing the number of authors (*p* = 0.047) and the number of investigated databases (*p* = 0.008).

Conversely, the number of articles included resulted as not correlated with the AMSTAR-2 score achieved by the study.

### 3.2. Multivariate Analysis

The multivariate analysis assessed the influence of qualitative and quantitative variables on the AMSTAR-2 score (Table 4). The year of publication (*p* < 0.001), the impact factor (*p* = 0.003), the number of research databases (*p* = 0.011), PRISMA statement declared (*p* = 0.003), the inclusion of a meta-analysis (*p* < 0.001), and the number of authors showed to be statistically significantly associated to the AMSTAR-2 score. A multi-country research team and the journal impact factor did not show a statistical significance for an increased AMSTAR-2 score.

The R² from the multivariate analysis shows that the AMSTAR-2 score can be predicted by the inclusion of the parameters by up to 23%.

Table 5 shows in which areas the findings of the multivariate analysis are similar to what was found using all the SRs (Table 4). It is clear that the most frequent significant independent variables associated to the AMSTAR-2 score found for each area were year of publication, number of research databases and meta-analysis.

## 4. Discussion

The main objective of this study was to define the scientific articles’ quality and evaluate how qualitative or quantitative variables could influence the AMSTAR-2 score. 

This study shows that number of authors, number of research databases, declared and applied PRISMA statement and meta-analysis inclusion increase the AMSTAR-2 score. Conversely, a multi-country research team and the journal impact factor did not significantly affect the AMSTAR-2 score.

An evident relationship between AMSTAR-2 score and the study impact factor score was found, and this is in agreement with the results found by Cheung et al. (Chinese Medicine) [27], McGregor et al. (pain management) [28], and in disagreement with Pauletto et al. (dentistry) [26] and Chow et al. (gynecology and obstetrics) [30]. In our study, an increase in the impact factor was associated with a higher score of the AMSTAR-2, suggesting that occupational medicine journals with a higher impact factor are more likely to publish systematic reviews with a higher methodological quality, as suggested by Saha et al., who found that usually researchers consider impact factor as a reasonable indicator of study quality [36].

Moreover, our study also found a significant and positive association between the AMSTAR-2 score and the number of authors in the study, and again, this is in agreement with the results found by Cheung et al. (Chinese Medicine) [27] and in disagreement with Chow et al. [30]. These findings could be linked to the so-called phenomenon of “knowledge diffusion”. As suggested by Tahamtan and coll. [37], by increasing the number of authors, a study is capable of increasing its faculty representation and the attention it receives, providing an increase in the credibility as well as the theoretical quality of evidence. 

In this study, an improvement over time of the methodological quality was found, as well as the association between the PRISMA statement declaration in the systematic reviews, the number of databases used and the AMSTAR score. This is in agreement with McGregor et al. (pain management) [28], Yuan et al. (oncologic surgery) [29], Yuan et al. (cosmetic surgery) [31], Aran et al. (psychiatry) [34], Ross et al. (pain management) [35], and in disagreement with Chow et al. (gynecology and obstetrics) [30]. This fact could be related, as suggested by Shamseer et al. [38], to the adoption of reporting guidelines for systematic reviews, endorsement of complete and transparent reporting by higher impact journals and increased attention to methodological quality.

Additionally, in this study, the AMSTAR-2 score was better in SRs with a meta-analysis than in SRs without a meta-analysis, and this could be due to the fact that the meta-analyses are useful tools for summarizing research evidence, as they can give an overall panorama on a specific research question.

Finally, we found some differences in the independent variables associated with methodological quality of the SRs in different occupational health areas. However, this result must be considered with care due to the lower number of studies included in the subgroup analyses. 

The first strength of this study is in the originality: similar studies could be found in the scientific literature, but they do not consider the AMSTAR-2 score compared with other qualitative and quantitative variables. A 2014 study about the impact of reporting guidelines on public health journals in Europe examined STROBE (Strengthening the Reporting of Observational Studies in Epidemiology), CONSORT (Consolidated Standards of Reporting Trials) and PRISMA and reported a strong heterogeneity in the application of guideline statements and suggested a common agreement among journals regarding research-reporting methodologies to improve the quality of PH (Public Health) research publishing [39]. A recent study analyzed the inter-rater reliability (IRR) of the AMSTAR-2 score and the ROBIS (Risk Of Bias In Systematic Reviews) scale considering individual domains and overall methodological quality/risk of bias of systematic reviews, the concurrent validity of the tools and the time required to apply them [40,41,42]. A second strength of this study is represented by the fact that the data extraction was performed independently by two authors, hence conferring robustness to the results. A multivariate regression model was then implemented in order to simultaneously analyze several variables.

The weaknesses of this study are the restriction to English language studies and the small number of investigated journals. In fact, only the 14 best occupational medicine journals were included, hence determining a quality bias for extracted data. Another possible limitation could be related to the fact that this search has not accounted for Cochrane reviews on this topic. However, we focused our attention on occupational health journals.

## 5. Conclusions

In conclusion, this study suggests that the rating of some variables such as number of authors, declared and applied PRISMA, number of research databases and meta-analysis inclusion, could be considered for an article quality analysis, with a predictivity in terms of R^2^ of 23% to define the AMSTAR-2 score. This screening could help for a fast identification of a good review or meta-analysis or, on the other hand, suggest some important elements that can have a positive association with the methodological quality of an SR. Future studies should analyze more journals without applying language restrictions and target a broader period of publication in order to furnish more robust evidence for results.

## Figures and Tables

**Figure 1 ijerph-20-01644-f001:**
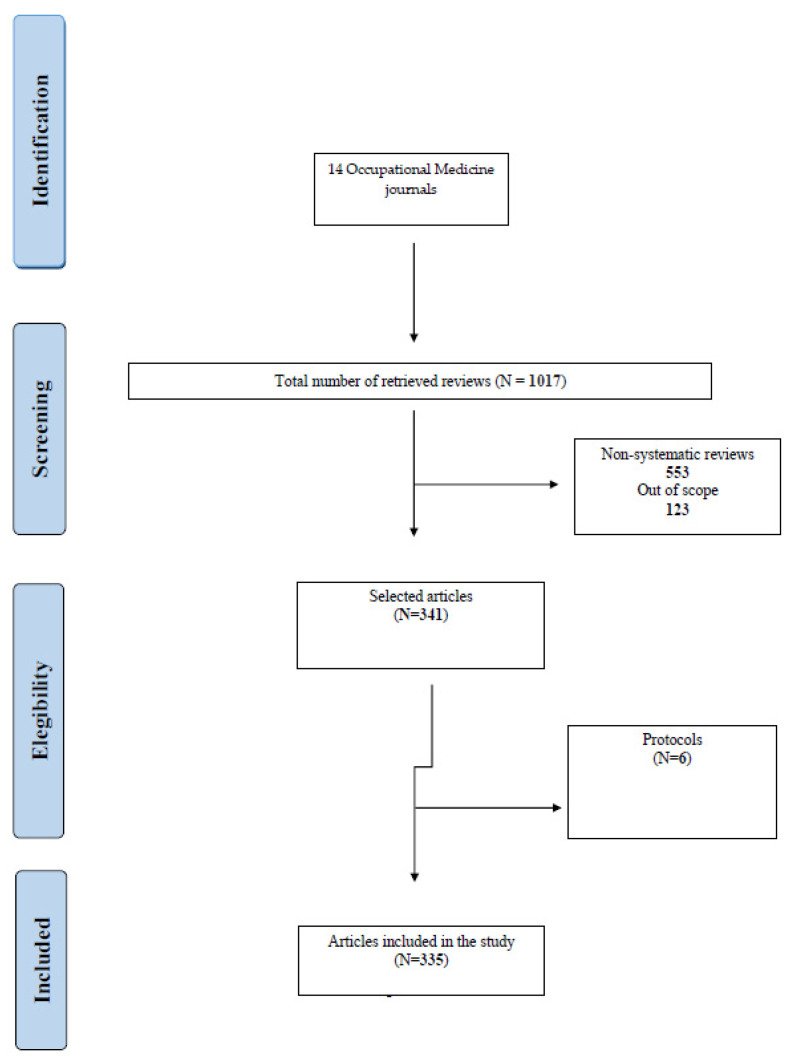
PRISMA flow-chart diagram of study selection.

**Figure 2 ijerph-20-01644-f002:**
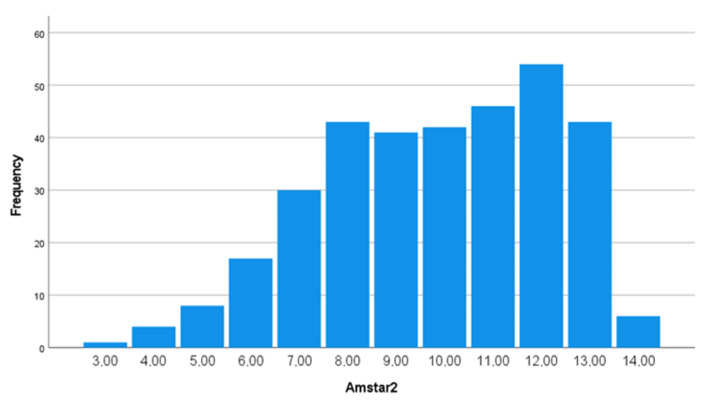
Histogram of the AMSTAR-2 score Frequency Distribution.

**Figure 3 ijerph-20-01644-f003:**
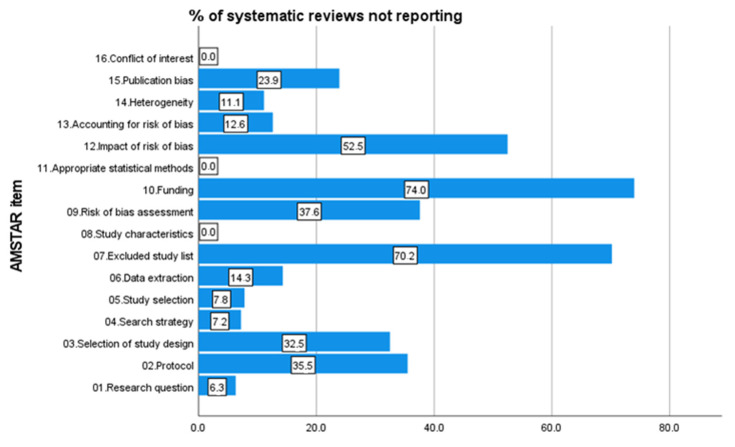
Bar chart of the percentages of systematic reviews not reporting the 16 items of the AMSTAR-2.

**Table 1 ijerph-20-01644-t001:** Occupational health areas covered by the systematic reviews.

Occupational Health Area	N	%
Toxicology	44	13.2
Health promotion and intervention	37	11.0
Environmental exposure and climate changes	36	10.7
Mental and neurological health	32	9.6
Musculoskeletal disorders	32	9.6
Stress and burnout	23	6.9
Workplace injury and Violence	21	6.3
Non-communicable diseases	18	5.4
Shift work	14	4.2
Return to work	12	3.6
Vaccination and infectious diseases	11	3.3
Healthcare workers	9	2.7
Asbestos	8	2.4
Ergonomy	7	2.1
Respiratory disease	6	1.8
Sickness absence	6	1.8
Tobacco smoking	5	1.5
Physical risk (ionizing and non-ionizing radiations, noise, vibration)	5	1.5
Precarious work and Unemployment	5	1.5
Other	4	1.2

**Table 2 ijerph-20-01644-t002:** Univariate Analysis of the AMSTAR-2 score vs. Qualitative Variables.

Qualitative Variables		AMSTAR-2 Score Mean (SD)	*p* Value
Multi-country Research Team	NO	9.88 (2.47)	0.762
	YES	9.79 (1.99)	
Prisma Statement Declared	NO	9.26 (2.53)	<0.001
	YES	10.35 (2.11)	
Prisma Statement Applied	NO	9.01 (2.61)	0.002
	YES	10.08 (2.25)	
Meta-Analysis	NO	9.38 (2.38)	<0.001
	YES	10.44 (2.23)	

**Table 3 ijerph-20-01644-t003:** Pearson’s correlation analysis of AMSTAR-2 score vs. quantitative variables.

Variables	r	*p* Value
Impact Factor	0.176	0.001
Publication year	0.316	<0.001
N. authors	0.109	0.047
N. of Research Database used	0.145	0.008
N. Articles Included	0.083	0.135

**Table 4 ijerph-20-01644-t004:** Multivariate regression model of AMSTAR-2 score as outcome.

Covariates	β	*p*
Impact Factor	0.154	0.003
Year of publication	0.274	<0.001
N. of Research Databases	0.128	0.011
Prisma Statement Declared (YES/NO *)	0.153	0.003
Meta-Analysis (YES/NO *)	0.211	<0.001
N. of Authors	0.111	0.046
R^2^ = 0.226		

* Reference group.

**Table 5 ijerph-20-01644-t005:** Independent variables present in the multivariate analysis for each occupational health area.

	Occupational Health Area					
Independent Variable	Toxicology	Health Promotion and Intervention	Environmental Exposure and Climate Changes	Mental and Neurological Health	Musculoskeletal Disorders	Stress and Burnout
Impact Factor		x				x
Year of publication	x		x	x		
N. of Research Databases	x	x		x		
Prisma Statement Declared (YES/NO *)			x		x	
Meta-Analysis (YES/NO *)	x		x			x
N. of Authors						

* Reference group.

## Data Availability

Data supporting reported results can be asked to the Authors.

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
