# Peer review of "What Are the Determinants of the Quality of Systematic Reviews in the International Journals of Occupational Medicine? A Methodological Study Review of Published Literature"

_ijerph, 2023, doi:10.3390/ijerph20021644_

Round 1

Reviewer 1 Report (Previous Reviewer 2)

It is important to assess the quality of systematic reviews in the field of occupational medicine. The authors addressed this issue. The introduction, materials and methods, and results are okay.

It is unfortunate that the discussion is scant and superficial. The present contents are as follows:

-         The first paragraph merely reiterated the study aim.

-         The second paragraph merely provides a summary of the findings.

-         In the third and fourth paragraphs, the authors showed the strengths and weaknesses of their study.

That is, they did not discuss the potential mechanisms by which impact factor, year of publication, the numbers of research databases used, authors, and articles included, presence of a PRISMA statement declaration, and employment of meta-analysis were associated with AMSTAR-2 scores. Nor they discussed why factors associated with AMSTAR-2 scores differed across occupational health areas. The authors must discuss these points in more detail.

Author Response

Reviewer 2 Report (New Reviewer)

Thank you for the opportunity to review this manuscript. The idea is an interesting and important one for occupational medicine. However, there are several important methodological concerns that must be addressed prior to publication.

The introduction provides a great deal of detail about reporting guidelines which are not relevant to the methodological quality of systematic reviews. There is a smaller section on AMSTAR 2 which is relevant but is only one tool, some additional background about the development of AMSTAR and descriptions of other tools for methodological quality of reviews (eg. ROBIS, SIGN) is warranted. This is quite important because the quality of reporting and methodological quality should not be confused.

The method of determining the sources of the reviews seems reasonable as does the eligibility criteria. However, be careful you have not “analyzed” the journals you have only searched for reviews in them. It is not clear whether this search will have accounted for Cochrane reviews on this topic which should be noted and indicated as a limitation if these systematic reviews were not consistently accounted for.

While there is a list of data items extracted, there is no justification for how these items could related to methodological quality. Why would nationality or number of authors be a determinant? An important step in a manuscript about determinants is to provide background from the literature on why variables would be associated with methodological quality. In addition, further description or operationalization of some of the items is required for the reader: eg what does ‘application’ of PRISMA mean? Another important issue is that AMSTAR2 is an instrument that explicitly states that it should not be scored and yet the authors have constructed a score. This may be reasonable but should be clearly justified and explained. There is also no information about partial scores which are a part of the AMSTAR2 or why the authors have ignored this aspect of the instrument. Given that the instrument was not designed to be scored there is also justification required for the analysis conducted i.e., simple correlation and regression. More clarity about the analysis is also required and some of the language of causation should be changed to focus on association throughout the manuscript (influence, affect).

The first lines of the results are confusing as the first line states there were 858 reviews and the second lines states there were 1057 articles with no explanation for the difference in these numbers. At least the second line agrees with the PRISMA figure. Again, given that AMSTAR2 was not meant to be scored, presenting the mean, SD, and median may not be meaningful. Justification and explanation for the scoring approach in the methods would help to rectify this issue.

The section in results about univariate and bivariate analysis is not clear especially the sentence about the letter “p” as Table 2 clearly indicates P value. The results suggest that the variables chosen were those that were available but there is no reason provided why many of these would be associated with methodological quality. The methods should clearly state how the variables were chosen. It is not clear what the value of Table 5 is to the manuscript as this was not described in the methods.

IN the discussion, once again, the quality of reporting is described but does not seem relevant to the results of the study per se. Please avoid causal language (predictive power, influence etc.). As analyzed and written there are unfortunately many more weaknesses in the study conducted that require revision. Key among them is that there is little information given as to why the variables were chosen, the scoring of the AMSTAR2 instrument and how association was analyzed.

Round 2

Reviewer 1 Report (Previous Reviewer 2)

I found the authors improved the discussion.

Line 294: "Saha and coll.," would be a misspelling.

Author Response

Reviewer 2 Report (New Reviewer)

Thanks to the authors for submitting a revised manuscript. I feel that most of my concerns have been addressed, save for the need to justify the independent variables considered in the analysis.

Introduction: The added text addresses my concern about the focus on reporting standards rather than methodological quality of systematic reviews. However I suggest removing some of the text about PRISMA as it is not the focus of the study. Specifically removing the description of all of the various extensions and simply stating that they exist and including the references so that the reader can go to them if desired.

Methods: The statement that other research has included the variables in the list presented is not sufficient. There are two reasons for this, 1) the references supporting that statement do not examine the association between SR quality and these variables, and 2) the current manuscript does not suggest why these may be associated. It is important to provide some logic for why these variables may be linked to the quality of the reviews.

There is no reason that I can think of why many of the variables listed would be linked to methodological quality especially the name of the author, name of journal (which may be highly correlated to impact factor – this should be checked and reported for all variables as well). I do see that the authors have provided some text in the discussion indicating some reasons for the associations found, however I am for the most part not convinced (see below).

In addition the variables need to be explained further or operationally defined – I still don’t know what “effective use of PRISMA” means, how was this determined and is a dichotomous score the best way to represent this? There should be text in the methods explaining the variables in some detail but it may also be possible to create a supplementary table that describes each variable and the reason for including. This would allow for less text in the methods as it would be a summary of the table.

I am unclear what the added sentence “Finally, independent variables present in the multivariate analysis for each occupational health area were indicated for making a comparison between them.” means.

Results: The addition of figure 3 is useful in my opinion.

Discussion: The added text describing the associations found is useful but I am not certain that the study findings support some of the statements. For example, AMSTAR2 score and impact factor – is it that authors equate impact factor with quality or could it be that journals with higher impact factors demand higher quality prior to publishing? For number of authors I am not sure how attention received and credibility are factors, these aspects are important ‘after’ publication but quality of the research is important prior to as well as after publication. Improvement over time makes sense. While I agree that meta-analyses are useful for summarizing findings, it is not clear how that is related to methodological quality.

The added text is a start but should be revised to explain how these factors could be associated with methodological quality. It should also reflect on the methods and why these variables were chosen in the first place. This requires careful consideration and should have been completed a priori. Stating that other research has included or explored the variable is not sufficient. At present the added discussion text seems to focus on knowledge translation elements rather than methodological quality.

Overall there are some improvements that can be built upon before publication. I would also suggest a review to address some English language/grammar issues.

Author Response

This manuscript is a resubmission of an earlier submission. The following is a list of the peer review reports and author responses from that submission.

Round 1

Reviewer 1 Report

From my point of view, this paper addresses an interesting issue, because due to the increase in systematic review and meta-analysis studies, there can always be doubts as to whether they are carried out with methodological quality.

I consider that the research has been well conducted and its limitations have been indicated at the end of the text, with which I agree. Likewise, the results, discussion and conclusions are also consistent with the approaches and objectives defended in the research.

In my opinion, the following aspects could be modified in order to improve the publication of this research:

1. I think there is an error in the introduction when it indicates the years of the review, it appears from 2014 to 2018, when throughout the document it appears from 2014 to 2021.

2. From my point of view, the flow chart should not be in the results section but in the methodology section, for example, when it talks about the eligibility criteria.

Reviewer 2 Report

As the authors proposed, it is important to assess the quality of systematic reviews in the field of occupational medicine.

The authors completely rely on the AMSTAR to evaluate the quality of systematic reviews. However, there is no explanation or discussion of the AMSTAR in the manuscript like the authors made the mistake to delete it.

-        I recommend the authors introduce the AMSTAR as well as the PRISMA in the Introduction.

-        Materials and Methods: (1) The AMSTAR items must be presented.; (2) Please explain who and how many researchers scored the AMSTAR. When two or more researchers independently scored for a certain article, the scores might have differed by the scorer. Please indicate what the authors did for the case.

-        I would like the authors to discuss whether the AMSTAR was the only and the most appropriate tool for the present study. The AMSTAR was originally developed to critically appraise systematic reviews of randomized trials. I do not believe that all systematic reviews the authors examined in their study were randomized trials. I suppose that the AMSTAR 2 was a better tool for the present study since it is designed to evaluate systematic reviews that include not only randomized but also non-randomized trials.

-        The authors appear to refer to Reference no. 24 as evidence of the AMSTAR. But, it is not quoted in the text.

Occupational medicine consists of various areas, from the toxicology of chemical materials to the psychiatric and mental health of workers. I would like to know what areas the authors covered in the present study. I also would like to know whether the present findings were consistent in every area.

As a supplementary file, the authors need to make a list of research papers that they employed for critical appraisal in the present study.

Reviewer 3 Report

Thanks for the opportunity to review the manuscript "What are the determinants of the quality of systematic reviews in the international journals of Occupational Medicine? A scoping review of published literature." The authors efforts are highly appreciated. However, there are some critical methodological concerns in the paper which need to be addressed.

The study is described as a "scoping review". This is not appropriate. A scoping review is an approach for systematically mapping evidence related to a broader topic of interest. The type of study that the authors have undertaken may be in line with a methodological study design. 

AMSTAR checklist was used in assessing the quality of systematic reviews and meta analyses. However, AMSTAR tool is designed ONLY for appraising the quality of systematic reviews of randomised controlled trials. In this study, the authors state that "This research study included all the systematic reviews and meta-analyses published 60 from the 14 most impacted occupational medicine journals from January 2014 to December 61 2021." So, clearly the systematic reviews and meta analyses were not examined to identify those that ONLY included randomised controlled trials. The systematic reviews and meta analyses included in the study may have very well been conducted on non-randomised studies. In such a case, the latest version of the AMSTAR tool (i.e., the AMSTAR 2 tool) must have been used. The AMSTAR 2 tool was specifically developed for appraising the quality of systematic reviews and meta analyses including randomised and non-randomised studies. Also, the developers of AMSTAR highly recommend against generating an "overall score" (as done in this study, 0-11) from the checklist, as the checklist is primarily meant to be used descriptively. These are critical methodological concerns for this study. It is highly recommended that the authors re-undertake the study based on the above recommendations.

Also, some suggestions to consider when writing the future manuscript: No description of the AMSTAR tool was provided either in the Introduction or Methods sections (not even the full form was mentioned anywhere in the paper). Most importantly, what are the various domains on the checklist and why are they significant? Since this tool is critical to this study, some overview of the tool required. 

Finally, the authors are advised to pay a close attention to English language, particularly spellings, vocabulary, and syntax.